# Characterization and Evaluation of Resistance to Powdery Mildew of Wheat–*Aegilops geniculata* Roth 7M^g^ (7A) Alien Disomic Substitution Line W16998

**DOI:** 10.3390/ijms21051861

**Published:** 2020-03-09

**Authors:** Yajuan Wang, Deyu Long, Yanzhen Wang, Changyou Wang, Xinlun Liu, Hong Zhang, Zengrong Tian, Chunhuan Chen, Wanquan Ji

**Affiliations:** 1College of Agronomy, Northwest A&F University, Yangling 712100, China; wangyj7604@163.com (Y.W.); longdeyu2018@sina.cn (D.L.); wangyanzhen9605@163.com (Y.W.); chywang2004@126.com (C.W.); Liuxxlun@126.com (X.L.); zhangh1129@nwafu.edu.cn (H.Z.); tian.zr@163.com (Z.T.); chchch8898@163.com (C.C.); 2State Key Laboratory of Crop Stress Biology for Arid Areas, Yangling 712100, China; 3Shaanxi Research Station of Crop Gene Resources and Germplasm Enhancement, Ministry of Agriculture, Yangling 712100, China

**Keywords:** wheat, *Aegilops geniculata* Roth, molecular cytogenetics, alien disomic substitution line, powdery mildew resistance.

## Abstract

*Aegilops geniculata* Roth has been used as a donor of disease-resistance genes, to enrich the gene pool for wheat (*Triticum aestivum*) improvement through distant hybridization. In this study, the wheat–*Ae. geniculata* alien disomic substitution line W16998 was obtained from the BC_1_F_8_ progeny of a cross between the common wheat ‘Chinese Spring’ (CS) and *Ae. geniculata* Roth (serial number: SY159//CS). This line was identified using cytogenetic techniques, analysis of genomic in situ hybridization (GISH), functional molecular markers (Expressed sequence tag-sequence-tagged site (EST–STS) and PCR-based landmark unique gene (PLUG), fluorescence in situ hybridization (FISH), sequential fluorescence in situ hybridization–genomic in situ hybridization (sequential FISH–GISH), and assessment of agronomic traits and powdery mildew resistance. During the anaphase of meiosis, these were evenly distributed on both sides of the equatorial plate, and they exhibited high cytological stability during the meiotic metaphase and anaphase. GISH analysis indicated that W16998 contained a pair of *Ae. geniculata* alien chromosomes and 40 common wheat chromosomes. One EST–STS marker and seven PLUG marker results showed that the introduced chromosomes of *Ae. geniculata* belonged to homoeologous group 7. Nullisomic–tetrasomic analyses suggested that the common wheat chromosome, 7A, was absent in W16998. FISH and sequential FISH–GISH analyses confirmed that the introduced *Ae. geniculata* chromosome was 7M^g^. Therefore, W16998 was a wheat–*Ae. geniculata* 7M^g^ (7A) alien disomic substitution line. Inoculation of isolate E09 (*Blumeria graminis* f. sp. *tritici*) in the seedling stage showed that SY159 and W16998 were resistant to powdery mildew, indeed nearly immune, whereas CS was highly susceptible. Compared to CS, W16998 exhibited increased grain weight and more spikelets, and a greater number of superior agronomic traits. Consequently, W16998 was potentially useful. Germplasms transfer new disease-resistance genes and prominent agronomic traits into common wheat, giving the latter some fine properties for breeding.

## 1. Introduction

Given ongoing population growth and conflicting resource demands, improvement of wheat (*Triticum aestivum* L.) yield is increasingly challenging. Loss of genetic diversity not only limits further improvement of quality and yield, but also increases the vulnerability of wheat to abiotic and biotic stresses. Powdery mildew caused by *Blumeria graminis* (DC.) Speer f. sp. *tritici* Em. Marchal (*Bgt*) is the most destructive wheat disease in the world [1]. The pathogen can attack all above-ground wheat parts, including spikes, leaves and stems. The consequence of using pesticides in large quantities is that the environment becomes polluted and costs are increased, so the breeding of resistant cultivars is the safest, and environmentally effective approach to prevent epidemics of the disease [2]. Accurate evaluation and effective utilization of resistant germplasms are prerequisites for the development of resistant cultivars. This is necessary to improve wheat disease resistance, enable screening of large germplasm resources, and enable the transfer and polymerization of resistance genes. It is an increasingly important approach for enriching common wheat with beneficial genes derived from related species by distant hybridization [3,4,5]. Many powdery mildew resistance genes are derived from wild relatives of wheat, and have been introduced to common wheat by additions, substitutions, and translocations of chromosomes. For example, *Pm21* and *Pm55* are derived from *Haynaldia villosa* Schur [6,7], and *Pm40* is derived from *Elytrigia intermedia* (Host) Nevski [8]. *Pm12*, *Pm32*, and *Pm53* are derived from *Aegilops speltoides* Tausch [9,10,11]. *Pm13* is derived from *Ae. longissima* Schweinf. and Muschl. [12], *Pm57* is derived from *Ae. searsii* Feldman and Kislev [13], and *Pm43* is derived from *Thinopyrum intermedium* (Host) Barkworth and D.R.Dewey [14]. Thus, the introduction of resistance genes from wild relatives is imperative and effective approach for broadening the genetic background of wheat.

*Ae. geniculata* Roth (ovate goatgrass; syn. *Ae. ovata* L. *pro parte*, 2n = 4*x* = 28, U^g^U^g^M^g^M^g^), a wild relative of annual wheat, contains many excellent traits for wheat breeding, such as high grain protein content, disease resistance and pest resistance [15,16,17,18], high grain iron and zinc content, early maturity [19], drought and heat adaption [15], and salt tolerance [20]. *Ae. geniculata* is a useful genetic resource for the improvement of cultivated bread (or common) wheat, because it is highly cross-compatible with common wheat [21]. A complete set of wheat–*Ae. geniculata* alien disomic addition lines was developed by Friebe et al. [22]. In 2007, Kuraparthy discovered two new disease-resistant genes in 5M^g^ of *Ae. geniculata*: stripe rust resistance gene *Yr40* and leaf rust resistance gene *Lr57* [23]. In 2002, Zeller found powdery mildew resistance gene *Pm29* in *Ae. geniculata*, and proved it is located on 7D chromosome*. Ae. geniculata* has broad potential for genetic improvements in wheat breeding [24].

In order to utilize the *Ae. geniculate* resistance genes in wheat, distant hybridization between the common wheat ‘Chinese Spring’ (CS) and *Ae. geniculata* SY159 has been carried out since 2009, backcrossing in F_1_ using CS, a series of wheat *Ae. geniculata* hybrids, including alien disomic addition line NA0973-5-4-1-2-9-1 [4], and some unrecognized hybrids. Among wheat–*Ae. geniculate* derivative lines, a new wheat–*Ae. geniculate* chromosome substitution line W16998 exhibited high resistance to powdery mildew in wheat-growing regions. In this study, the objectives were: (1) to determine the chromosome numbers and genomic composition of the alien chromosomes using cytogenetic analysis and GISH (genomic in situ hybridization) identification of line W16998; (2) to determine the homologous group relationships of introduced *Ae. geniculata* chromosomes, using molecular markers, FISH (fluorescence in situ hybridization) and sequential FISH–GISH analysis, and nullisomic–tetrasomic analysis of CS; (3) to investigate agronomic character and powdery mildew resistance.

## 2. Results

### 2.1. Cytological Characterization

The results of cytological observation indicated that the root tip chromosome numbers were 2n = 42 (Figure 1A). We looked at 120 root tip cells: 113 cells had 42 chromosomes, comprising 94.17% of the total number of observations. A total of 61 pollen mother cells were observed: 56 pollen mother cells had a value of 2n = 21II in meiotic metaphase (Figure 1B). No trivalents or quadrivalents were observed at meiotic metaphase I. No laggard chromosomes were observed at meiotic anaphase I and chromosomes were distributed evenly on both sides of the equatorial plate (Figure 1C). Therefore, the line W16998 exhibited high cytological stability.

To determine the numbers of alien chromosomes of the derived line, W16998 was analyzed by GISH, using SY159 genomic DNA as the probe and CS as barrier. The result was that, there, two chromosomes exhibited strong green hybridization signals in the root tip cells (Figure 2). These results indicate that W16998 comprised two *Ae. geniculata* chromosomes and 40 common wheat chromosomes.

### 2.2. Molecular Marker Analysis

In order to clarify the relationship of the *Ae. geniculata* homologous chromosome group to the alien disomic substitution line W16998, we analyzed the distribution of 300 EST–STS and PLUG markers on seven homologous groups of wheat. The data reveal that 112 primers (37.3%) had polymorphisms between CS and SY159. Eight primers (7.1%) had distinctive bands amplified in SY159, W16998 and CS. These eight markers (*BE637663*-7AL 7BL 7DL (EST–STS), *TNAC1868*-7AL 7BL 7DL, *TNAC1782*-7AS 7BS 7DS, *TNAC1829*-7AS 7BS 7DS, *TNAC1845*-7AL 7BL 7DL, *TNAC1888*-7AL 7BL 7DL, *TNAC1929*-7AL 7BL 7DL, and *TNAC1941*-7AS 7BS 7DS (PLUG)) were mapped to the seventh homologous group. This is evidence that these markers were specific markers of *Ae. geniculata* chromosomes in W16998 (Table 1, Figure 3). This shows that *Ae. geniculata* 7U^g^ or 7M^g^ chromosomes were introduced into common wheat CS. In other words, these chromosomal components had *Ae. geniculata* 7U^g^ or 7M^g^ chromosomes.

### 2.3. Nullisomic–Tetrasomic Analysis

These marker results indicate that two alien chromosomes were homoeologous with the wheat group 7 chromosomes. On the basis of amplification of the nullisomic–tetrasomic lines of CS, the homoeologous group 7 PLUG markers amplified specific bands for wheat chromosomes 7A, 7B and 7D. Ultimately, two PLUG markers, *TNAC1868-Taql* and *TNAC1941-Taql*, clearly amplified fragments of chromosomes 7A, 7B, and 7D in CS, and *Ae. geniculata*-specific bands were detected in W16998 (Figure 4). These results verify that the *Ae. geniculata* chromosomes introduced into W16998 belonged to the seventh homologous group, and the chromosome 7A-specific band was absent in W16998.

### 2.4. FISH and Sequential FISH–GISH Analysis

To determine the identity of the wheat chromosomes replaced by *Ae. geniculata* chromosomes in W16998, FISH and sequential FISH–GISH analyses were performed. Oligo-pSc 119.2 (green signal) and Oligo-pTa535 (red signal), two oligonucleotide probes, were able to clearly distinguish the 42 wheat chromosomes simultaneously [25]. Compared with the FISH karyotype of CS (Figure 5A) and W16998 (Figure 5C), it was shown that one pair of 7A wheat chromosomes was absent in W16998. These results accorded with the results of nulli-tetrasomic analysis. One pair of chromosomes in W16998 showed special signals. One pair of the alien chromosomes carried red signal (pTa535) on both ends, in contrast with the FISH analysis of the 7M^g^ alien disomic addition lines NA0973-5-4-1-2-9-1 (Figure 5B). This pair of specific chromosomes in W16998 (Figure 5C) is highly similar to the 7M^g^ chromosome (Figure 5B). The results of sequential FISH–GISH analysis show that this pair of specific chromosomes in W16998 has strong signs of *Ae. geniculate*. To sum up, based on FISH analysis combined with nulli-terasomic analysis, and on functional molecular marker screening in combination with sequential FISH-GISH observation, W16998 was designated a wheat–*Ae. geniculata* 7M^g^ (7A) alien disomic substitution line.

Interestingly, an additional strong green signal (Oligo-pSc119.2) appeared on the end of the short arm of chromosome 1A in W16998 (Figure 5C). This was different from the FISH karyotype of CS, with NA0973-5-4-1-2-9-1 of 7M^g^ alien disomic addition lines on the end of the short arm of chromosome 1A. Other common wheats have no green signal (Oligo-pSc119.2). We have marked it with white arrows (Figure 5A–C).

### 2.5. Evaluation of Powdery Mildew Resistance and Agronomic Traits

In 2018 and 2019, at the seedling stage, we planted 30 plants each of CS, SY159 and W16998. These were inoculated using the *Bgt* isolate E09 of powdery mildew, and their susceptibility compared to that of ‘Shaanyou 225’. At 15 days after inoculation, when the powdery mildew spores were fully developed on the leaves of the ‘Shaanyou 225’, the results showed that the SY159 and W16998 plants were nearly immune to powdery mildew. In contrast, CS was highly susceptible (Figure 6E). These results suggested that the wheat–*Ae. geniculata* 7M^g^ alien disomic substitution line inherited a high degree of powdery mildew resistance from *Ae. geniculata*.

The average spikes of W16998 plants were bulkier than those of CS, and the spikes bore long awns similar to those of the parent *Ae. geniculata* SY159 (Figure 6A). On average, the tillers of W16998 were higher than those of CS, but shorter than those of SY159. The average number of kernels per spike and the 1000-grain weight of W16998 were significantly higher than in CS and SY159. The number of kernels per spike was 50 grains. The 1000-grain weight was 40 g (Table 2), which was significantly higher, according to Duncan’s multiple range test (*p* < 0.01).

## 3. Discussion

Chromosome engineering research is important for wheat breeders, to develop new disease-resistant germplasms and broaden the genetic base [17,26,27,28]. Friebe et al. [22] reported that the *Ae. geniculata* (accession TA2899) was crossed (as the male parent) with CS, to obtain a complete set of wheat*–Ae. geniculata* chromosome addition lines, where TA7667 was an *Ae. geniculata* 7M^g^ alien disomic addition line. Wang et al. [18] reported the NA0973-5-4-1-2-9-1 of a wheat–*Ae. geniculata* 7M^g^ alien disomic addition line (*Ae. geniculata* accession: SY159) after inoculation with *Bgt* isolate E09: NA0973-5-4-1-2-9-1 showed almost immune to powdery mildew, whereas TA7667 was susceptible. Meanwhile, Zeller et al. [24] conducted chromosomal mapping of the powdery mildew resistance gene *Pm29* in the common mildew-resistant wheat line Pova, derived from the wheat ‘Poros’–*Ae. geniculata* (accession TA2899) alien addition line. The *Bgt* races used in the study were collected from different parts of Europe and selected from single spore progenies. On the basis of the afore-mentioned information, W16998 contains a novel powdery mildew resistance gene different to *Pm*29. These results suggest that the 7M^g^ chromosomes of *Ae. geniculata* contain valuable powdery mildew resistance genes for the genetic improvement of wheat. However, it is important and necessary to continue to study and map powdery mildew resistance genes from *Ae. geniculata* 7M^g^ line W16998.

It is an important part of genetic wheat improvement to introduce beneficial genes from other species into common wheat by means of distant hybridization. The techniques of FISH and GISH analysis can not only determine the constitution and number of chromosomes, but can also detect alien chromosomes or introgressive segments introduced into common wheat’s genetic background efficiently and accurately [29,30]. FISH and GISH have high sensitivity, are simple to program, generate obvious contrast, and can detect multiple probes at the same time. These characteristics have quickly turned them into mainstream techniques for in situ hybridization. For instance, Mariyana Georgieva reported that line 55 (1–57) contained 42 wheat chromosomes and six *Th. intermedium* pairs, including two S and one J^S^ pairs, while line H95 contained 44 wheat chromosomes and four *Thinopyrum* chromosome pairs—including one J chromosome and three S pairs—using GISH. FISH analysis detected a null (1D)-tetrasomy (1B) in 55(1–57) and a 6B tetrasomy in H95 [31]. In addition, various forms of functional molecular markers are also highly effective, display good stability, are easy to operate and are low-cost. Using molecular markers can not only reliably identify alien chromosomes or introgressive fragments in wheat’s genetic background, but can also be used to define the homoeologous group relationships of alien chromosomes from related species [32]. For example, EST and PLUG markers have been used to identify the homoeologous group relationship of alien chromosomes’ widespread availability [33]. In this study, GISH analysis showed that W16998 incorporated two alien chromosomes from *Ae. geniculata*. Using functional molecular markers, nullisomic–tetrasomic analysis FISH and FISH–GISH methods showed that the two alien *Ae. geniculata* chromosomes in W16998 belonged to homoeologous group 7 (Figure 3), the common wheat chromosome, 7A, was replaced in W16998. This substitution might be the result of a chain of events, such as asymmetrical bivalent formation and subsequent gametic selection. Chromosomes 7A and 7M^g^ of W16998 have high homology; the produced chromosomes exchange and recombine during the separation of progeny. According to previously published studies, chromosome structure recombination, chromosome constitution variation, and genomic changes can trigger occasionally during the process of interspecific hybridization or allopolyploidization. This is referred to as “genome shock” [34,35]. In previous papers, the presence of pSc119.2 sites was variable [36,37]. Interestingly, structural differences in chromosomes between CS and W16998 were detected, based on differences in the signal of the Oligo-pSc119.2 probe on the short arm of chromosome 1A (Figure 5). This observation indicates that wheat chromosomes may experience extensive restructuring and structural alterations, as well as wheat–wheat translocations, when common wheat is crossed with relatively wild species [22,25,34]. This might be due to an undetected translocation, or to an increase in the copy number at the target spot [38], but further study is necessary for validation.

Kernels per spike are one of the three factors determining wheat yield. It is very important to create new germplasm with multi-grain characteristics through distant hybridization. Breeding for large spikes by increasing the number of kernels per spike is an option for improving the yield potential of wheat [39,40]. *Haynaldia*, *Elytrigia*, *Leymus, Aegilops*, *Secale*, and others, have all been successfully hybridized with common wheat: the offspring germplasm contains large spikes, higher numbers of spikelets, more kernels per spike and more florets [41,42]. Wu et al. [41] reported introgression from the 6P chromosome of *Agropyron cristatum*, conferring increased numbers of florets and kernels. This was the first reported transfer of an *A. cristatum* chromosome to common wheat, and enhanced the floret and kernel numbers of the wheat. Friebe et al. [22] showed that spikes in the 4M^g^ alien disomic addition line contain more florets than usual. In the present study, W16998 showed a relatively greater number of superior agronomic traits compared to its parents. In particular, the W16998 line produced longer spikes, more tillers, more spikelets per spike, more kernels per spikelet, and more kernels per spike, and 1000-grain weight was increased (Figure 6, Table 2). Consequently, the wheat–*Ae. geniculata* 7M^g^ (7A) alien disomic substitution line W16998, on account of its desirable agronomic characteristics and strong powdery mildew resistance, may be useful as a novel donor source for wheat chromosome engineering breeding.

## 4. Materials and Methods

### 4.1. Materials

*Ae. geniculata* (No. SY159) was provided by Dr. Lihui Li and Dr. Xinming Yang of the Institute of Crop Sciences, Chinese Academy of Agricultural Sciences, Beijing, China. This genotype is resistant or almost resistant to mixed *Bgt* isolates in northern China. The ‘Chinese Spring’ (CS) wheat and SY159 began a process of distant hybridization in 2008. The individual plants in F_1_ were backcrossed with CS. The disomic substitution line W16998 was isolated from the BC_1_F_8_ progeny, which exhibited resistance powdery mildew in the field, and somatic cell chromosome number 2n = 42, in 2018 and 2019. CS and its nulli–tetrasomic lines (CSN7AT7B, CSN7AT7D, CSN7BT7A, and CSN7DT7A) were used in a molecular cytogenetic analysis to determine the chromosomal location. NA0973-5-4-1-2-9-1 (CS-AEGEN DA 7M^g^), with 7M^g^ chromosomes from *Ae. geniculate,* was used as a control [18]. The wheat ‘Shaanyou 225′ was used as a mildew-susceptible control in tests of powdery mildew resistance. The *Bgt* isolate E09 is a widely prevalent isolate, and was used to test powdery mildew resistance [43]. All of the above-mentioned materials were obtained from the College of Agronomy, Northwest A&F University, China.

### 4.2. Cytological Observation

The seeds were immersed in water on a wet petri dish with filter paper at room temperature for 1 day, until they germinated and become white. Following germination, they were placed in darkness in a constant-temperature incubator at 23 °C. When the root tips grew to 2–4 cm, they were excised, put into a centrifuge tube, and pretreated with nitrous oxide for two hours. They were then fixed in 95% acetic acid for 10 min and placed in 70% ethanol. The root tips were treated with 1% pectinase and 2% cellulose at 37 °C for 1 h, using a production process described by Han et al. [44]. At the appropriate stage of development, young panicles were excised and put into a tube of ethanol–chloroform–acetic acid solution (6:3:1, *v*/*v*/*v*) for one week at 25–30 °C. Anthers were extracted and squashed on a slide in 1% acetocarmine. The number of root tip chromosome and pollen mother cell chromosome pairs was observed with a light microscope and photographed.

### 4.3. GISH, FISH and Sequential FISH–GISH

The genomic DNA of *Ae. geniculata* SY159 was used as the probe for GISH, and the genomic DNA of CS was used as blocking DNA. The *Ae. geniculata* genomic DNA was labeled with fluorescein-12-dUTP. The GISH procedure was performed as described by Fu [29] and Pei [45], with minor modifications. The oligonucleotide probes Oligo-pTa535 (red) and Oligo-pSc119.2 (green) (Shanghai Invitrogen Biotechnology Co. Ltd., Shanghai, China) were used for FISH and sequential FISH–GISH analysis, as described by Tang et al. [25]. The results of in situ hybridization and fluorescent signals were viewed and photographed with an Olympus BX-43 microscope equipped with a DP80 camera [4].

### 4.4. EST–STS and PLUG Markers Analysis

Expressed sequence tag–sequence-tagged site (EST–STS) markers were selected from the Wheat Haplotype Polymorphisms Website (http://wheat.pw.usda.gov/SNP/new/pcr_primers.shtml). PCR-based landmark unique gene (PLUG) markers were synthesized by AuGCT DNA-SYN Biotechnology Co., Ltd., Beijing, China [31,46]. These markers come from homoeologous groups 1 to 7 in wheat chromosomes. The PCR amplification and electrophoresis procedures were as described previously [4,31].

### 4.5. Powdery Mildew Resistance and Agronomic Trait Evaluation

CS, SY159, W16998, and ‘Shaanyou 225′ were used to evaluate to powdery mildew resistance at the seedling stage in the greenhouse. The infection type was recorded for 30 plants, as previously described [47,48]. At two weeks after inoculation with *Bgt* isolate E09, when the powdery mildew spores of ‘Shaanyou 225′ were fully infected, the plants were investigated and infection type (IT) was evaluated. Survey results were recorded using a 0–4 scale. Plants with IT of 0–2 were judged resistant to powdery mildew, whereas plants with IT of 3–4 were judged susceptible to powdery mildew.

The morphological traits of line W16998, and its parents CS and SY159, were assessed at the physiological maturity stage in 2018 and 2019, in the field. Ten plants were selected randomly and their agronomic traits were recorded, including plant height, tillering, spke length, number of kernels per spike, number of spikelets per spike, number of kernels per spikelet, 1000-grain weight, and awnedness [4]. All agronomic trait data were analyzed using Duncan’s multiple range test, and significant differences were found (*p* < 0.01).

## 5. Conclusions

In this work, we have characterized and identified line W16998 from the BC_1_F_8_ progeny of crossing between ‘Chinese Spring’ (CS) wheat and *Ae. geniculata* SY159, using cytology, GISH analysis, EST–STS, PLUG analysis, FISH analysis, sequential FISH–GISH analysis, and disease resistance assessment. The results showed that line W16998 was a wheat–*Ae. geniculata* 7M^g^ (7A) alien disomic substitution line, with substitution conferring powdery mildew resistance. This substitution line is potentially useful. Germplasms transfer new disease-resistance genes and determine prominent agronomic traits for wheat breeding and chromosome engineering.

## Figures and Tables

**Figure 1 ijms-21-01861-f001:**
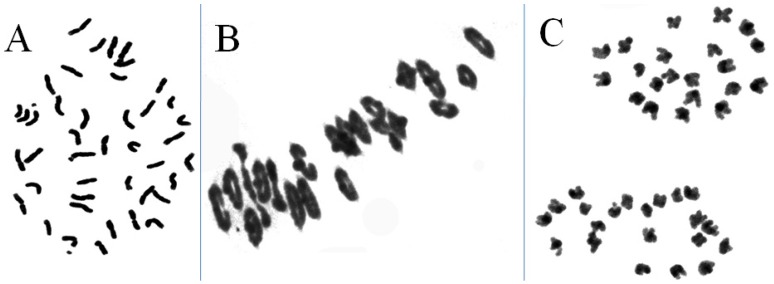
Cytogenetic analysis of W16998. (**A**). root tip cells at mitotic metaphase, 2n = 42. (**B**). pollen mother cell chromosomal configurations at meiotic metaphase, 2n = 21П. (**C**). pollen mother cell chromosomal configurations at anaphase I, 2n = 21 + 21.

**Figure 2 ijms-21-01861-f002:**
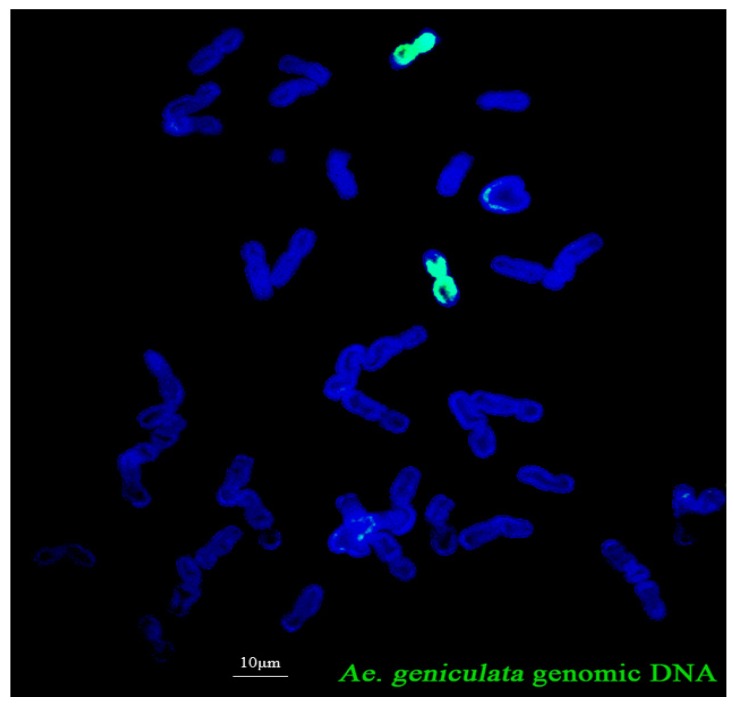
GISH (genomic in situ hybridization) analysis of W16998, using *Ae. geniculata* SY159 genomic DNA as a probe (green) and CS (Chinese Spring) genomic DNA as a blocker on root tip metaphase I. Chromosomes were counterstained using DAPI (blue).

**Figure 3 ijms-21-01861-f003:**
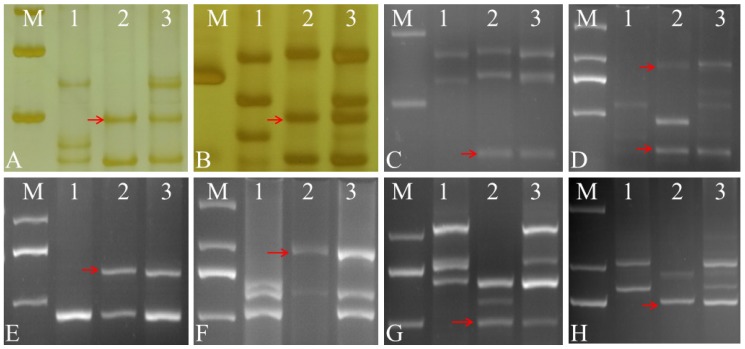
EST–STS and PLUG functional molecular marker analysis of W16998. The red arrows indicate the SY159-specific bands. (M). DL2000 (2 kb DNA ladder). (1). CS. (2). SY159. (3). W16998. (**A**). *BE637663*. (**B**). *TANC1868-TaqI*. (**C**). *TNAC1782-HaeII*. (**D**). *TNAC1829-TaqI*. (**E**). *TNAC1845-TaqI*. (**F**). *TNAC1888-TaqI*. (**G**). *TNAC1929-TaqI*. (**H**). *TNAC1841-TaqI.*

**Figure 4 ijms-21-01861-f004:**
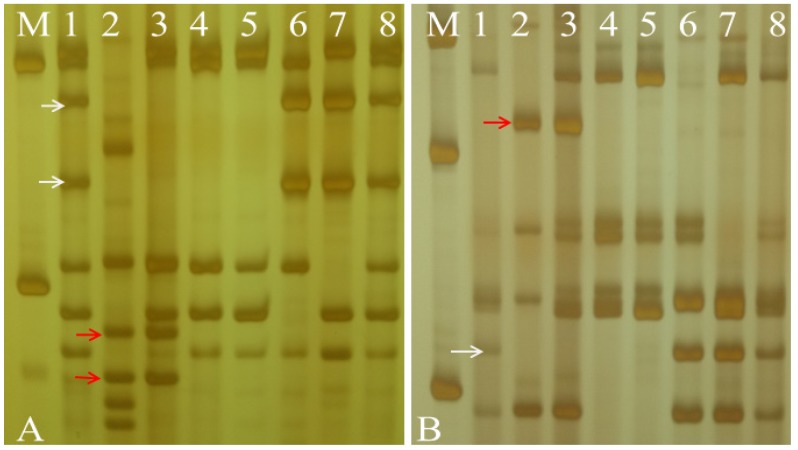
Nullisomic-tetrasomic analysis of W16998. The red arrows indicate the SY159-specific bands. The white arrows indicate CS and the nullisomic-tetrasomic-specific bands. (M). DL2000. (1). CS. (2). SY159. (3). W16998. (4). CSN7AT7B. (5). CSN7AT7D. (6). CSN7BT7A. (7). CSN7DT7A. (8). CS. (**A**). *TNAC1868-TaqI*. (**B**). *TNAC1941-TaqI*.

**Figure 5 ijms-21-01861-f005:**
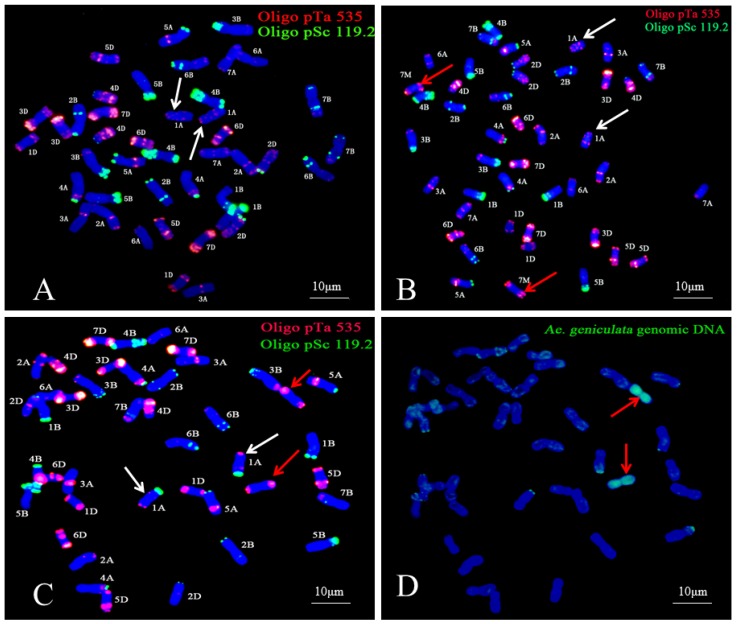
Karyotypes with the genomic composition variation of W16998, obtained using FISH and sequential FISH–GISH analyses. The probes for FISH were Oligo-pSc119.2 (green) and Oligo-pTa535 (red). The probe for sequential FISH–GISH was SY159 genomic DNA (green). The red arrows indicate *Ae. geniculata* chromosomes; the white arrows indicate structural variations in the chromosomes. (**A**). FISH of CS. (**B**). FISH of NA0973-5-4-1-2-9-1 (CS-AEGEN DA 7M^g^). (**C**). FISH of W16998. (**D**). GISH of W16998 in the same cell as (**C**).

**Figure 6 ijms-21-01861-f006:**
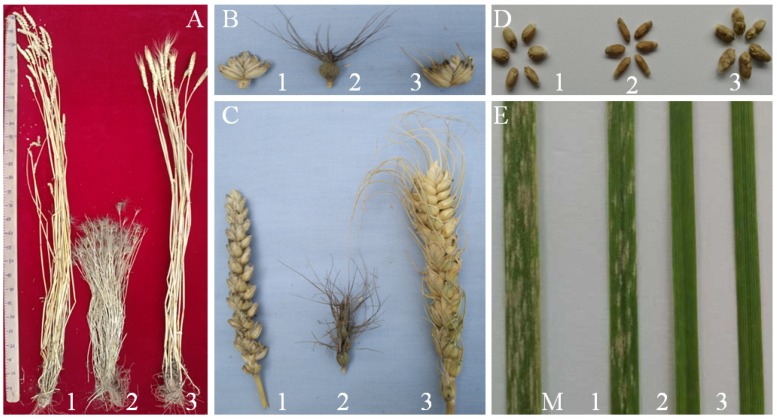
Morphological and powdery mildew reactions of W16998. (1) CS; (2) SY159; (3) W16998. (M) Shaanyou 225. (**A**). plants. (**B**). florets. (**C**). spikes. (**D**). kernels. (**E**). symptoms in response to inoculation with E09 at the seedling stages.

**Table 1 ijms-21-01861-t001:** Expressed sequence tag-sequence-tagged site (EST–STS) and PCR-based landmark unique gene (PLUG) marker list for W16998.

Marker	Type	Primer (5’-3’)	Location	Geltype/Restrictionenzyme	Tm °C/t (h)
*BE637663*	EST-SSR	F: ACTGTTGCTTCGCTCCAAGT R: GTTCCATTTCCGATGTGCTC	7AL 7BL 7DL	8% non-denaturing polyacrylamide gel/-	60/-
*TNAC1868*	PLUG	F: CTCCGCCTTCATCGGAAA R: CCGTTCTGCTTCAGGATCTC	7AL 7BL 7DL	8% non-denaturing polyacrylamide gel/-	60/-
*TNAC1782*	PLUG	F: TCACTGAACAGCCTAGACATGG R: ATTCGCAGACCGCATCTATC	7AS 7BS 7DS	2% agarose gel/*TaqI*/*HaeIII*	60/2 or 37/2
*TNAC1829*	PLUG	F: GCCACTTCCTCCCTCCTC R: GTCGGTCCTCCAGTATCAGC	7AL 7BL 7DL	2% agarose gel/*TaqI*	60/2
*TNAC1845*	PLUG	F: AATGAACAGCTTGCTTTCTGC R: CAGATGCTCTGGATTTCATGG	7AL 7BL 7DL	2% agarose gel/*TaqI*	60/2
*TNAC1888*	PLUG	F: AGGGATGTGTTGGAGCTGTTA R: CACAGTGACCTTCTGCTCCTT	7AL 7BL 7DL	2% agarose gel/*TaqI*	60/2
*TNAC1929*	PLUG	F: GCACCAGAAGGTTCAGTAGCA R: ATCTGTCAGCAGGGCACACT	7AS 7BS 7DS	2% agarose gel/*TaqI*	60/2
*TNAC1941*	PLUG	F: AATGATCCTGACAAGGTGCAG R: GTAGCGATGGCATCCAGAGA	7AS 7BS 7DS	2% agarose gel/*TaqI*	60/2

- represent no data.

**Table 2 ijms-21-01861-t002:** Analysis of the agronomic traits of W16998 and its parents (CS, SY159).

Materials	Tillering	Plant Height (cm)	Spike Length (cm)	Spikelets/Spike	Kernels/Spikelet	Kernels/Spike	Thousand Kenel Weight (g)	Awnedness
CS	13 ± 4	130 ± 5	9.0 ± 0.3	20 ± 2	4 ± 1	40 ± 4	30 ± 0.5	awnless
SY159	75 ± 5	65 ± 5	2.8 ± 0.4	3 ± 1	3 ± 1	10 ± 2	18 ± 2.0	Long awn
W16998	15 ± 5	120 ± 5	11.0 ± 0.5 **	23 ± 2	5 ± 2	50 ± 5 **	40 ± 1.0 **	Long awn

** Indicates significant differences between the substitution line W16998 and wheat parent CS (*p* < 0.01).

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
