# Peer review of "Characterization and Evaluation of Resistance to Powdery Mildew of Wheat–Aegilops geniculata Roth 7Mg (7A) Alien Disomic Substitution Line W16998"

_ijms, 2020, doi:10.3390/ijms21051861_

Round 1

Reviewer 1 Report

This paper represents the development of wheat-Aegilops chromosome translocated lines to breed a resistance gene to powdery mildew. They identified the chromosome by FISH and marker analysis. I think the research objective, results and discussion are enough for a full paper. New variety W16998 is a potential one to improve the trait of the powdery mildew resistance and the gene identification relating the trait in the future. This will contribute to the academic knowledge for the wheat breeders.

I put on the correction and suggestion in the pdf document. There are lot of miner mis-text. I wish the authors to correct them by spellchecking before their submitting manuscript again.

Reviewer 2 Report

The paper is quite interesting and reports novel information. Nevertheless it needs to be improved before it can be accepted for pubblication. In general the quality of the English is acceptable, but there are sentences, here and there, that are very difficult to understand, or are truncated thus missing a logical meaning, or contain grammar fails. I suggest the authors to carefully ceck the text before re-submitting. Ahead some points to be fixed or improved.

  • Line 22. A disomic substitution should have 21 bivalents, as also stated in the results section
  • Same line, please correct line name W16998 not W1699 8
  • A description of how the line W16998 was obtained from the disomic addition line is missing. Lines 78-91 describe the procedure of checking somatic chromosome numbers and metaphase I meiotic pairing, but no strategy is reported. The authors MUST describe the strategy not only the methods
  • Line 83-84. Please consider that the correct stage is "meiotic metaphase I"  not meiosis I, since the meiosis is the entire reductional process comprising two cell division cycles
  • lines 149-153 The authors describe the appearance of a strong pSc119.2 signal on chromosome 1A. This might depend on an undetected translocation or to an increase of copy number at target spot (e.g. see Guo et al. (2019) Mol Breeding 39, 133). No mention of this is given in the discussion. Could the authors please comment?
  • Lines 169-179. The authors report of a 8-year period of selfing of the line W16998. Do they refer to a period after the identification of its chromosomal structure or to a period preceding it? In brief, did they identify the disomic substitution more than 8 years ago? Unclear
  • In general the discussion is reporting unecessary information and can be sensibly shortened without losing arguments. Please avoid redundant sentences that make reading less confortable
  • In detail lines 225-231 make little sense. The presence of the alien chromosome itself does not imply the substitution of a somatic wheat chromosome. The substitution might be the result of a chain of events such as asymmetrical bivalent formation and subsequent gametic selection, or lots of other possibilities. If the authors describe the selection strategy they followed, they can also speculate on the possible events leading to chromosome 7A substitution
  • Lines 263-264. Do not understand. Please describe in a better way
  • Lines 271-279. Please improve the methods description

The authors did a very good job and it is their duty to better decribe it in order to fully expose the quality of their hard work.

Round 2

Reviewer 2 Report

The revised version of the paper fulfills my previous requests. The paper is now fit for publication provided that the authors check the english language and perform an accurate spell check

I will suggest "minor revisions" only for what regards text editing

This manuscript is a resubmission of an earlier submission. The following is a list of the peer review reports and author responses from that submission.